Machine learning based estimation of field-scale daily, high resolution, multi-depth soil moisture for the Western and Midwestern United States

http://orcid.org/0000-0002-8250-5381 Xia Yushu 1 yxia@woodwellclimate.org
Watts Jennifer D. 1
Machmuller Megan B. 2
http://orcid.org/0000-0002-3215-1706 Sanderman Jonathan 1
1 Woodwell Climate Research Center , Falmouth, Massachusetts , United States
2 Department of Soil and Crop Sciences, Colorado State University , Fort Collins, Colorado , United States
Karabiniuk Mykola
Electronic publication date: 2022 Nov 4
Publication date: 2022
Volume: 10
Electronic Location ID: e14275
Received 2022 Aug 4; Accepted 2022 Sep 29
Copyright: © 2022 Xia et al.
Copyright year: 2022
Copyright holder: Xia et al.
License: This is an open access article distributed under the terms of the Creative Commons Attribution License, which permits unrestricted use, distribution, reproduction and adaptation in any medium and for any purpose provided that it is properly attributed. For attribution, the original author(s), title, publication source (PeerJ) and either DOI or URL of the article must be cited.
License URL: https://creativecommons.org/licenses/by/4.0/

Keywords: Soil moisture downscaling, Field scale, Soil climate analysis network (SCAN), U.S. Climate Reference Network (USCRN), North American Land Data Assimilation System (NLDAS), Digital soil mapping, Environmental covariates, Spiking, Grassland, Remote sensing

Funding: Conscience Bay Research, LLC. and Woodwell Fund for Climate Solutions Rangeland Tracking and Management This work was supported by the Rangeland Tracking and Management project funded by Conscience Bay Research, LLC and the Woodwell Fund for Climate Solutions and Conscience. The funders had no role in study design, data collection and analysis, decision to publish, or preparation of the manuscript.

==============================
Background

High-resolution soil moisture estimates are critical for planning water management and assessing environmental quality. In-situ measurements alone are too costly to support the spatial and temporal resolutions needed for water management. Recent efforts have combined calibration data with machine learning algorithms to fill the gap where high resolution moisture estimates are lacking at the field scale. This study aimed to provide calibrated soil moisture models and methodology for generating gridded estimates of soil moisture at multiple depths, according to user-defined temporal periods, spatial resolution and extent.

Methods

We applied nearly one million national library soil moisture records from over 100 sites, spanning the U.S. Midwest and West, to build Quantile Random Forest (QRF) calibration models. The QRF models were built on covariates including soil moisture estimates from North American Land Data Assimilation System (NLDAS), soil properties, climate variables, digital elevation models, and remote sensing-derived indices. We also explored an alternative approach that adopted a regionalized calibration dataset for the Western U.S. The broad-scale QRF models were independently validated according to sampling depths, land cover type, and observation period. We then explored the model performance improved with local samples used for spiking. Finally, the QRF models were applied to estimate soil moisture at the field scale where evaluation was carried out to check estimated temporal and spatial patterns.

Results

The broad-scale QRF model showed moderate performance (R2 = 0.53, RMSE = 0.078 m3/m3) when data points from all depth layers (up to 100 cm) were considered for an independent validation. Elevation, NLDAS-derived moisture, soil properties, and sampling depth were ranked as the most important covariates. The best model performance was observed for forest and pasture sites (R2 > 0.5; RMSE < 0.09 m3/m3), followed by grassland and cropland (R2 > 0.4; RMSE < 0.11 m3/m3). Model performance decreased with sampling depths and was slightly lower during the winter months. Spiking the national QRF model with local samples improved model performance by reducing the RMSE to less than 0.05 m3/m3 for grassland sites. At the field scale, model estimates illustrated more accurate temporal trends for surface than subsurface soil layers. Model estimated spatial patterns need to be further improved and validated with management data.

Conclusions

The model accuracy for top 0–20 cm soil depth (R2 > 0.5, RMSE < 0.08 m3/m3) showed promise for adopting the methodology for soil moisture monitoring. The success of spiking the national model with local samples showed the need to collect multi-year high frequency (e.g., hourly) sensor-based field measurements to improve estimates of soil moisture for a longer time period. Future work should improve model performance for deeper depths with additional hydraulic properties and use of locally-selected calibration datasets.

Introduction

The monitoring of soil moisture is vital for maintaining ecosystem services such as plant and soil health (Lowery et al., 1997; Jiang et al., 2021), erosion and pollution control (Jebellie, Prasher & Clemente, 1996; Wei, Zhang & Wang, 2007), biodiversity, and habitat protection (Sylvain & Wall, 2011). Because of the close association between soil moisture and ecosystem processes including C and nutrient cycling, water and solution movement, and soil biological activities, soil moisture is often used as a key model input to estimate crop and forage productivity, C sequestration potentials, greenhouse gas emissions, and environmental risks (Dominati, Patterson & Mackay, 2010; Grassini et al., 2015; Kuang et al., 2019; Udawatta & Gantzer, 2022). Accurate estimation of soil moisture is therefore considered necessary to support ecosystem assessments and management decisions.

Due to the high variability in weather conditions, soil hydraulic properties, and water movement (Bormann & Klaassen, 2008; Dari et al., 2019), dense spatiotemporal sampling is required to delineate the spatial variability and temporal dynamics of soil moisture when relying solely on in-situ measurements which include direct, lab-based gravimetric method, and indirect, field sensor-based methods that rely on neutron probes, electromagnetic techniques, cosmic-ray neutrons, optical reflectance, tensiometers, and thermal dissipation (Stafford, 1988; Walker, Willgoose & Kalma, 2004; Dorigo et al., 2011). Field and lab-based in-situ measurements can be costly and time-consuming, making it difficult to characterize soil moisture at higher resolutions needed to inform management. Consequently, soil moisture has been commonly estimated through geostatistical interpolation, proximal and remote sensing (RS), and empirical and process-based modeling (Robinson et al., 2008; Srivastava, 2017; Kathuria, Mohanty & Katzfuss, 2019; Babaeian et al., 2019). In particular, a covariate-based digital soil mapping (DSM) framework (McBratney, Mendonça Santos & Minasny, 2003), which considers soil properties to be a function of soil-forming factors, has been applied at various scales to empirically derive soil moisture estimates where machine learning (ML) models were built upon crucial soil, climate, topographic, and biotic covariates (Ahmad, Kalra & Stephen, 2010; Martínez-Murillo, Hueso-González & Ruiz-Sinoga, 2017; Zhang et al., 2020; Adab et al., 2020). Advancements in RS and increased availability of survey-based datasets has improved covariate datasets in recent years which would enable soil moisture to be empirically estimated at finer resolutions with higher accuracy.

Even though ML-based models are known for simplicity and predictive power, they are often criticized for a lack of transparency. It is well acknowledged that ML-based models should be thoroughly evaluated with independent validation datasets to ensure reasonable model selection and transferability (Lamichhane, Kumar & Wilson, 2019; Khaledian & Miller, 2020). Since models are usually only transferable within the inference space delineated by the calibration dataset (Miller et al., 2016), it is crucial to use large datasets spanning multiple climatic, soil, and vegetation gradients for model training in order to cover the data range needed for prediction. For soil moisture estimates within the contiguous U.S., national databases with wide spatial coverages (Table 1) such as Soil Climate Analysis Network (SCAN) (Schaefer, Cosh & Jackson, 2007) and U.S. Climate Reference Network (USCRN) (Diamond et al., 2013) that compiled a large number of in-situ measurements, present an important opportunity for model training and evaluation.

Table 1 Representative broad-scale soil moisture networks based on in-situ measurements with data available for the contiguous U.S.

Networka	Multiple networks	No. sites	Spatial distribution	Temporal coverage	Temporal resolution	Depths	Sensor method	Quality control	Other measured covariatesb	
SCAN	No	>200	Uneven and mostly from agricultural lands	Since the 90s	Hourly	5, 10, 20, 51, 102 cm	Dielectric constant	Yes	ppt, soil T, SR, RH, WS and WD, air T, air P, soil pedon info	
SNOTEL	No	>450	Uneven and mostly from the western states	Since the 30s	Hourly to daily	5, 10, 20, 50, 100 cm	Dielectric constant	Yes	ppt, soil T, SR, RH, WS and WD, air T, air P	
NASMD	Yes	>1,800 in U.S. and Canada	Uneven but were harmonized to gridded scale	Vary by network	Vary by network and harmonized to daily	Vary by network but harmonized to standard depths	Vary by network	Automated procedure established	Land cover, texture, BD, hydraulic conductivity, elevation	
Ameriflux	No	>400 sites	Uneven	Vary by site but mostly after 2000	Mostly half-hourly	Vary by site	Vary by site	Vary by sites	LULC, ppt, soil T, VPD, SR, RH, WS and WD, air T, air P, C fluxes	
USCRN	No	>100	Relatively even	Since 2000	Hourly and daily	5, 10, 20, 50, 100 cm	Dielectric constant	Yes	ppt, soil T, SR, RH, WS, air T	
COSMOS	No	>50	Uneven	Mostly after 2010	Hourly	Estimated as effective measurement depths	Cosmic-ray neutrons	Standard procedure established	SOC	
ISMN	Yes	>1,400 globally	Uneven and more from the western states	Since the 50s	Vary by network and harmonized to hourly	Vary by network	Vary by network	Automated procedure established	LULC, ppt, soil T, climate classification	
Notes:

a Ameriflux (Baldocchi et al., 2001; Law, 2005); COSMOS, COsmic-ray Soil Moisture Observing System (Zreda et al., 2012); ISMN, International Soil Moisture Network (Dorigo et al., 2011); NASMD, North American Soil Moisture Database (Quiring et al., 2016); SCAN, Soil Climate Analysis Network (Schaefer, Cosh & Jackson, 2007); SNOTEL, SNOwpack TELemetry (Schaefer & Paetzold, 2000); USCRN, U.S. Climate Reference Network (Diamond et al., 2013).

b Air P, air pressure; air T, air temperature; BD, bulk density; ppt, precipitation; LULC, land use land cover type; RH, relative humidity; SOC, soil organic carbon; soil T, soil temperature; SR, solar radiation; VPD, vapor pressure deficit; WD, wind direction; WS, wind speed.

The national soil moisture datasets can serve as baseline maps for land managers but are usually low in spatial resolution (>10 km) and unevenly distributed in space, which requires further data interpolation and harmonization (Table 1). Previous efforts have also relied on interpreting soil moisture directly from RS products (e.g., NASA’s Soil Moisture Active-Passive (SMAP) Mission), which can cover a wide spatial range with relatively fine temporal resolution but is still lacking the finer spatial resolution details needed for decision making (Moran et al., 2000; Hunt et al., 2003; Velpuri, Senay & Morisette, 2016). Combining high-quality RS datasets, modeling techniques, and in-situ soil moisture measurements from national databases through data fusion can help generate high resolution gridded soil moisture products needed for agricultural monitoring (Peng et al., 2017; Cosh et al., 2021); however, land managers are still faced with the difficulty of retrieving and processing data derived from RS products without spending extensive resources on obtaining advanced technical support and expert knowledge. Soil moisture monitoring is particularly challenging for rangeland management considering that ranches commonly encompass a great number of land cover types such as grasses, hay, pasture, and shrubs which are associated with high heterogeneity in soil moisture dynamics. Some ranch properties even have proportions of areas in forest, bare ground, and crops. Under rapidly changing climate conditions (Derner et al., 2018), there is a growing need to rethink grazing management for U.S. rangelands to adapt to and potentially help mitigate climate change, and provide positive ecosystem services such as C sequestration and food security (Schuman, Janzen & Herrick, 2002; Havstad et al., 2007; Briske et al., 2015), where accurate soil moisture information can be critical for drought monitoring and productivity estimates (Richardson & Everitt, 1992; Sohrabi et al., 2015).

A number of soil moisture products/models developed with data fusion methods may satisfy the need of users in the U.S. even though none of these products are specifically designed for rangeland and therefore may lack parameters specifically calibrated for different land use/land cover (LULC) types in ranch properties (Table S1). Moreover, the majority of the RS-driven models were focused only on the top soil layer (0–5 cm) probably due to the strength of RS products in predicting surface soil moisture but not root zone moisture. However, the latter is of great importance to plant distribution and growth is usually under-emphasized (Schnur, Xie & Wang, 2010). In addition, the current moisture products have limited spatial resolution ( ≥1 km) and/or temporal coverage (only available after 2015 because of their model reliance on Sentinel or SMAP products) that is needed for decision making based on spatially-explicit information and long-term records. Even with the limitations, some of the studies (Table S1) which have provided associated data products or codes, such as Adaptive High-Resolution Soil Moisture Map (AHRSMM) (Huang et al., 2020), PYthon Sentinel-1 Soil-Moisture Mapping Toolbox (PYSMM) (Greifeneder, Notarnicola & Wagner, 2021), and SMAP-Hydroblocks (SMAP-HB) (Vergopolan et al., 2021), are examples for a move towards developing automatic algorithms that are user-friendly and supported by background data processing. Built upon DSM and data fusion frameworks, this study aimed to provide models calibrated with the national databases to allow extraction of long-term and multi-depth soil moisture estimates at high spatial (e.g., 30 m) and temporal resolutions (daily). Model covariates were carefully selected to represent ecosystem processes and to capture the temporal range needed for soil moisture monitoring. We evaluated model performances in relation to calibration datasets and then provided an estimation of resources needed to generate soil moisture layers for site-based management.

Materials and Methods

Extract in-situ soil moisture measurements from national databases

For broad-scale model calibration and validation datasets, we extracted in-situ soil moisture measurements from SCAN (Schaefer, Cosh & Jackson, 2007) and USCRN (Diamond et al., 2013) because they have similar measurement depths and sensor types which makes it easier for data harmonization (Table 1). In addition, the datasets have been quality-controlled and have spatial and temporal coverages that are suited for soil moisture modeling for the contiguous U.S. In this study, we provide soil moisture estimates for the Western or the Midwestern U.S. states (Reeves & Mitchell, 2012; Spangler, Burchfield & Schumacher, 2020) so only stations that fall within the defined spatial domain were retained (Fig. 1).

Figure 1 Stations with soil moisture calibration datasets used in this study.

The stations were extracted from national soil moisture monitoring networks including Soil Climate Analysis Network (SCAN) (Schaefer, Cosh & Jackson, 2007) and U.S. Climate Reference Network (USCRN) (Diamond et al., 2013) and assigned with the dominant land use type during the past 20 years (2001–2019) based on the U.S. National Land Cover Database (Homer et al., 2004) dataset. The NLCD data layer is shown for the year 2011.

Data from the retained stations were further screened based on the following quality-control criteria adapted from the North American Soil Moisture Database (Liao et al., 2019): (1) records that did not report sampling depths were excluded; (2) records showing soil moisture below or equal to 0, or higher than 0.6 m3/m3 were excluded; (3) records were joined with soil temperature records to exclude frozen soils. Since our study is interested in modeling longer term records of soil moisture, we further selected data measured between 2002 and 2019. We then retained records measured from the surface up to 100 cm to include deeper depth records. The retained records measured from 5, 10, 20, 50, and 100 cm depths were then processed to represent 993,771 site-depth-date combinations from 120 stations. The major land use land cover (LULC) classes identified for the stations (Fig. 1) include shrubs (33%), grassland (23%), cropland (17%), and forest (17%) according to the National Land Cover Database (NLCD) data layer (Homer et al., 2004). The data extraction and processing steps were carried out in R (R Core Team, 2021). All codes for this and the following steps are provided in our GitHub repository (RCTM-soil-moisture) and are available through the link: https://github.com/xiayushu/RCTM-soil-moisture. Table S2 contains processing steps numbered to represent the corresponding scripts in the repository.

The coordinates corresponding to retained SCAN and USCRN stations were extracted and used as sampling points to create 90 m buffers for the subsequent extraction of Land surface model (LSM)-based soil moisture estimates and environmental covariates (Fig. 2).

Figure 2 Flowchart showing the datasets and processes for model calibration, validation, and prediction.

The national datasets used for model calibration and validation are Soil Climate Analysis Network (SCAN) and U.S. Climate Reference Network (USCRN). Covariates used in this study include North American Land Data Assimilation System (NLDAS), Land Surface Temperature (LST), Normalized Difference Wetness Index (NDWI), Enhanced Vegetation Index (EVI), Gross Primary Productivity (GPP), soil texture, bulk density (BD), soil organic carbon (SOC), precipitation, temperature (Temp), vapor pressure deficit (VPD), digital elevation model derived variables and indices, land use land cover (LULC), and tree cover percentage.

Process LSM-based soil moisture estimates

We extracted LSM-based soil moisture estimates from the North American Land Data Assimilation System (NLDAS) database (Xia et al., 2012a) for the contiguous U.S. as the baseline soil moisture map for this study. The NLDAS hourly soil moisture is provided at a gridded scale (0.125 degrees) since 1979 that so far has the highest spatial resolution for the contiguous U.S. that is publicly available for the past 20 years. The NLDAS soil moisture is estimated with the Noah LSM which simulates soil freeze-thaw, heating-cooling, evaporation, transpiration, infiltration, and runoff processes based on model inputs including climate variables (precipitation, temperature, humidity, air pressure, wind speed, solar radiation) and surface parameters (vegetation type, green vegetation fraction, soil texture, roughness, albedo, slope factor) (Ek et al., 2003; Xia et al., 2012a). A list of URLs for available soil moisture estimates was downloaded from the EARTHDATA portal using the subset tools (Xia et al., 2012b). Then, Google Colaboratory (Google Inc., Menlo Park, CA, USA) was used to extract images from URLs corresponding to data from 2002 to 2019 before images representing three depth layers (0–10, 10–40, and 40–100 cm) were processed to daily averages and uploaded to the Google Earth Engine (GEE) (Gorelick et al., 2017) platform as an asset. Finally, point buffer-based averages of NLDAS-estimated soil moisture were extracted on a site-date-depth basis (Fig. 2), where we matched the median depth of the NLDAS layer with the sampling depth of observations from the national databases.

Extract and process environmental covariates

We selected covariates representing four classes including soil properties, climate conditions, biotic factors, and topography, extracted from digital elevation models (DEMs) (Table 2), because of their known importance for soil moisture status (Verstraeten, Veroustraete & Feyen, 2008; Vereecken et al., 2014). Our selected covariates largely overlapped with previous work but encompassed a wider range since most of the previous work only utilized two or three covariate classes, with climate and biotic factors being the most common choices (Table S1). In addition, our framework differed from previous efforts in that we considered covariates that are available for at least 20 years (since 2002) so that long-term moisture records can be simulated using our model.

Table 2 Data sources, variables, and their extraction criteria for building the soil moisture model.

Data type	Variable or covariatesa	Resolution	Temporal coverage	Extraction criteria	In GEE	Source and referenceb	
Spatial	Temporal	
In-situ observation	Soil moisture	Unevenly distributed in space	Hourly or daily	Since the 90s		No	SCAN and USCRN (Schaefer & Paetzold, 2000; Diamond et al., 2013)	
Land surface model	Soil moisture	0.125°	Hourly	Since 1979	By site, by depth, by date	No	NLDAS (Xia et al., 2012b)	
Soil properties	Soil texture, SOC, BD	100 m	NA	NA	By site, by depth	Yes	SoilGrid+ (Ramcharan et al., 2018)	
Climate	air T, ppt, VPD	1 km	Daily	1980–2021	By site, by date	Yes	DAYMET v4 (Thornton et al., 2020)	
4.6 km	Daily	Since 1981	By site, by date	Yes	PRISM (Daly et al., 2008)	
Biotic	LST, NDWI	1 km	Daily	Since 2000	By site, by date	Yes	MODIS (Huete, Didan & Leeuwen, 1999; Platnick et al., 2003)	
EVI	463 m	Daily	Since 2000	By site, by date	Yes	
GPP	250 m	Every 8 days	Since 2000	By site, by date	Yes	
Tree cover%	250 m	Annually	2000–2020	By site, by year	Yes	
LULC class	30 m	Every 1 to 5 years	Since 1992	By site, by year	Yes	NLCD (Homer et al., 2004)	
LULC%	30 m	Annually	Since 1984	By site, by year	Yes	RAP (Jones et al., 2018)	
Digital elevation model	Elevation, slope, aspect, vertical, horizontal, and mean curvatures	30 m	NA	NA	By site	Yes	SRTM (van Zyl, 2001)	
TWI	15 arc sec	By site	Yes	HydroSHEDS (Lehner, Verdin & Jarvis, 2008)	
Surface roughness	90 m	By site	Yes	Geomorpho90m (Amatulli et al., 2020)	
Notes:

a Air T, air temperature; BD, bulk density; EVI, Enhanced Vegetation Index; GPP, Gross Primary Productivity; LULC, land use land cover; LST, Land Surface Temperature; NDWI, Normalized Difference Wetness Index; ppt, precipitation; SOC, soil organic carbon; TWI, Topographic Wetness Index; VPD, vapor pressure deficit.

b DAYMET, Daily Surface Weather and Climatological Summaries; HydroSHEDS, Hydrological data and maps based on SHuttle Elevation Derivatives at multiple Scales; MODIS, Moderate Resolution Imaging Spectroradiometer; NLCD, National Land Cover Database; NLDAS, North American Land Data Assimilation System; SCAN, Soil Climate Analysis Network; SRTM, Shuttle Radar Topography Mission; PRISM, Parameter elevation Regression on Independent Slopes Model; RAP, Rangeland Analysis Platform; USCRN, U.S. Climate Reference Network.

Soil covariates including soil texture (clay and sand contents), soil organic C (SOC), and bulk density (BD) were extracted from the SoilGrid+ product which maps soil properties at multiple depths from the contiguous U.S. at a 100 m resolution (Ramcharan et al., 2018). We retained the data layers representing standard depths of 5, 15, 30, 60, and 100 cm. In order to match the estimates with the in-situ soil moisture records, the first and second, second and third, and third and fourth depth layers were averaged to create predictions that approximate soil properties at depths of 10, 20, and 50 cm, respectively. The extraction for soil covariates was carried out on a site-depth basis (Table 2).

We extracted climate covariates, including daily precipitation, air temperature, and vapor deficit pressure (VPD) from the DAYMET v4 product which is available at a spatial resolution of 1 km (Thornton et al., 2020). Since the DAYMET data is generally not up-to-date on GEE, we provided optional, supplemental code for using the Parameter elevation Regression on Independent Slopes Model (PRISM) product (Daly et al., 2008) which has a coarser spatial resolution (4.6 km) but longer temporal coverage. Our code was developed to extract climate covariates, and make soil moisture predictions, according to site locations and dates specified by the user. Accordingly, using PRISM in the model instead of DAYMET is only needed for soil moisture predictions beyond 2021 (Table 2).

Biotic covariates consist of RS-derived indices and LULC classes. We extracted Land Surface Temperature (LST), Normalized Different Wetness Index (NDWI), and Enhanced Vegetation Index (EVI) on a site-date basis from the Moderate Resolution Imaging Spectroradiometer (MODIS) (Huete, Didan & Leeuwen, 1999; Platnick et al., 2003) satellite imagery. The MODIS-derived Gross Primary Productivity (GPP) was available every 8 days at a spatial resolution of 250 m (Robinson et al., 2018); so the values were extracted based on data from the nearest available date (Table 2). We also used MODIS product (Townsend & DiMiceli, 2015) to extract tree cover % on a site-year basis. The LULC classes were extracted from the NLCD product (Homer et al., 2004) on a yearly basis where information from the closest year was used when data is not available for the specific year.

We extracted elevation, slope, and aspect from the Shuttle Radar Topography Mission (SRTM)-based DEMs. Further, the mean, vertical, and horizontal curvatures were processed from SRTM using the “TAGEE” package (Safanelli et al., 2020). The topographic wetness index (TWI) was extracted from Hydrological data and maps based on Shuttle Elevation Derivatives at multiple Scales (HydroSHEDS) (Lehner, Verdin & Jarvis, 2008) and the surface roughness was determined using the Geomorpho90m (Amatulli et al., 2020) product. The extraction for DEM-based topographic covariates was carried out on a site basis (Table 2).

The covariate extraction step was implemented with the High-Performance Computing (HPC) platforms (Table S2). Soil, LULC, and topographic covariates were extracted in GEE while biotic and climate covariates were extracted in Google Colaboratory to avoid computation time-out issues. Finally, separate covariate files were combined in R for subsequent quality control and model evaluation.

Quality-control and reduction of the calibration dataset

We screened the NLDAS-based estimates to exclude outliers (soil moisture estimates >0.6 m3/m3). GEE-based MODIS bitmasks were used to exclude pixels with lower quality caused by cloud cover, geometry problems, or emissivity issues. This step produced a dataset that only contains records associated with RS pixels labeled as “good quality”. This quality-controlled dataset contains 586,337 observations from 114 stations and will be referred to as the “full” dataset hereafter. We examined the Pearson correlation between the extracted modeling covariates and soil moisture observations for the full dataset (Table S3).

For LULC class, we used a rangeland-specific LULC product, the Rangeland Analysis Platform (RAP) (Jones et al., 2018) as an alternative modeling approach, to account for the land cover percentage of common vegetation types (annual herbs, perennial herbs, shrubs, trees, litter, and barren) in rangelands in addition to the NLCD product. In this case, the finer resolution tree cover% extracted from RAP (30 m) was used to replace MODIS-derived tree cover which is provided at a coarser resolution (250 m) (Table 2). The RAP layers are only available for the western U.S., so the quality-controlled dataset was further reduced. This step generated a reduced dataset with 500,119 observations from 94 stations which will be referred to as the “regionalized” dataset hereafter.

Model calibration and model validation

We first generated calibration datasets by randomly dividing the study sites into the calibration (70%) and validation (30%) sets. Since model performance can vary with sites used for independent validation, we carried out the random division five times to get different sets of calibration and validation sites for subsequent model building and evaluation. The Quantile Random Forest (QRF) models adapted from AHRSMM (Huang et al., 2020) were built with our calibration datasets in R using the “caret” and “quantregForest” packages (Kuhn, 2008). The QRF model (Meinshausen, 2006) combines the concept of both quartile regression (Koenker & Bassett, 1978) and random forest (RF) (Breiman, 2001). It therefore has the advantages of tree-based models including producing higher accuracy estimates and being less sensitive to model overfitting (Belgiu & Drăguţ, 2016; Hengl et al., 2018). However, QRF is different from the conventional RF model in that median instead of mean is used for point prediction. In particular, the QRF model gives non-parametric estimates for high-dimensional variables and is considered to be more consistent and less influenced by extreme values (Meinshausen, 2006; Vaysse & Lagacherie, 2017; Gyamerah, Ngare & Ikpe, 2020).

The QRF models for the full model were tested with different tree numbers (10, 15, 20, 25, 30, 35, 40, 45, and 50) and node sizes (5, 10, 15, and 20) to reduce overfitting, which is achieved by comparing model accuracy between the calibration and the validation dataset. The model was finally established using 15 trees and five nodes based on the 5-fold cross-validation results. To further examine the potential of model overfitting and the utility of different covariate classes, we also facilitated additional model runs using reduced covariate datasets as shown in Table S4.

Calibration models were built and validated for both the full and the regionalized datasets. Furthermore, separate models were built on subsets divided based on sampling depths, land use classes, and sampling periods (Table 3) to evaluate model efficiency associated with the use of different calibration datasets. Error metrics, including coefficient of determination (R2) (Eq. (1)), root mean square error (RMSE) (Eq. (2)), mean bias error (MBE) (Eq. (3)), and Residual Prediction Deviation (RPD) (Eq. (4)) were derived from independent sites held out for validation. Covariate importance was reported based on QRF’s increase in node purity measure in the calibration models.

Table 3 Summary statistics of the full and regionalized dataset divided by sampling depth, land cover classes, and month of the year.

Classes	Full dataseta	Regionalized datasetb	
Obs. (Frequency)	Range (m3/m3)	Mean ± stdev (m3/m3)	Median (m3/m3)	Obs. (Frequency)	Range (m3/m3)	Mean ± stdev (m3/m3)	Median (m3/m3)	
All	586,337 (100%)	0.001–0.600	0.201 ± 0.119	0.181	500,119 (100%)	0.001–0.600	0.182 ± 0.109	0.163	
Sampling depth	
5 cm	124,009 (21%)	0.001–0.596	0.156 ± 0.106	0.136	106,422 (21%)	0.001–0.596	0.137 ± 0.096	0.118	
10 cm	128,072 (22%)	0.001–0.597	0.192 ± 0.104	0.179	109,693 (22%)	0.001–0.597	0.177 ± 0.097	0.165	
20 cm	117,817 (20%)	0.001–0.599	0.213 ± 0.110	0.201	99,836 (20%)	0.001–0.599	0.197 ± 0.103	0.185	
50 cm	108,572 (19%)	0.001–0.600	0.225 ± 0.123	0.211	91,503 (18%)	0.001–0.600	0.207 ± 0.112	0.191	
100 cm	107,777 (18%)	0.001–0.600	0.226 ± 0.136	0.188	92,665 (19%)	0.001–0.600	0.201 ± 0.122	0.169	
Land cover class	
Grassland	145,166 (25%)	0.001–0.599	0.201 ± 0.108	0.188	145,166 (29%)	0.001–0.599	0.201 ± 0.108	0.201	
Cropland	133,636 (23%)	0.001–0.600	0.242 ± 0.126	0.229	105,640 (21%)	0.001–0.600	0.217 ± 0.119	0.205	
Forest	80,017 (14%)	0.001–0.585	0.229 ± 0.123	0.221	46,045 (9%)	0.001–0.577	0.181 ± 0.109	0.169	
Pasture	47,919 (8%)	0.001–0.600	0.264 ± 0.130	0.256	23,669 (5%)	0.001–0.600	0.238 ± 0.119	0.227	
Shrubs	177,816 (30%)	0.001–0.598	0.144 ± 0.085	0.129	177,816 (36%)	0.001–0.598	0.141 ± 0.085	0.129	
Month of the year	
1	20,324 (3.5%)	0.001–0.600	0.228 ± 0.125	0.207	16,576 (3%)	0.001–0.600	0.199 ± 0.113	0.180	
2	19,449 (3%)	0.001–0.599	0.228 ± 0.124	0.211	16,154 (3%)	0.001–0.599	0.202 ± 0.111	0.182	
3	33,708 (6%)	0.001–0.600	0.241 ± 0.121	0.236	28,531 (6%)	0.001–0.600	0.219 ± 0.111	0.213	
4	44,622 (8%)	0.001–0.600	0.238 ± 0.121	0.235	37,687 (8%)	0.001–0.600	0.218 ± 0.112	0.213	
5	51,472 (9%)	0.001–0.599	0.222 ±0.120	0.209	44,106 (9%)	0.001–0.599	0.204 ± 0.111	0.191	
6	70,984 (12%)	0.001–0.600	0.196 ± 0.117	0.176	62,588 (13%)	0.001–0.600	0.182 ± 0.109	0.163	
7	74,441 (13%)	0.001–0.600	0.184 ± 0.113	0.163	64,702 (13%)	0.001–0.600	0.171 ± 0.105	0.153	
8	77,699 (13%)	0.001–0.599	0.177 ± 0.112	0.156	67,583 (13%)	0.001–0.597	0.164 ± 0.104	0.147	
9	77,274 (13%)	0.001–0.599	0.181 ± 0.114	0.159	65,618 (13%)	0.001–0.598	0.164 ± 0.104	0.146	
10	59,761 (10%)	0.001–0.600	0.190 ± 0.113	0.170	50,701 (10%)	0.001–0.600	0.172 ± 0.104	0.153	
11	36,496 (6%)	0.001–0.600	0.201 ± 0.117	0.183	29,635 (6%)	0.001–0.600	0.173 ± 0.104	0.153	
12	20,107 (3.5%)	0.001–0.600	0.221 ± 0.126	0.202	16,238 (3%)	0.001–0.600	0.190 ± 0.113	0.169	
Notes:

a The full dataset contains records from Midwestern and Western U.S. states.

b The regionalized dataset contains records from Western U.S. states. The dataset was created according to the spatial domain of the Rangeland Analysis Platform (RAP) dataset.

(1) R2=1−∑i=1n⁡(yi−y^i)2∑i=1n⁡(yi−y¯)2

(2) RMSE=∑i=1n⁡(yi−y^i)2n

(3) MBE=∑i=1n⁡(yi−y^i)n

(4) RPD=SDvalRMSEPn/(n−1)

where n represents the number of samples, yi represents observed value of sample i, y^i represents the model predicted value of sample i, y¯ represents the mean of observations, SDval represents standard deviation of the validation dataset, and RMSEP represents the RMSE of the validation dataset.

An alternative model calibration scheme was carried out to test if model performance can be improved with the spiking strategy that incorporates local samples for model building. Specifically, the calibration dataset consists of both data from the calibration sites from the national database and local data from 10%, 30%, 50%, 70%, and 90% of the study years at the sites that were held out for validation. Model performance was then evaluated using data from the years that were not used for calibration at each validation site. Summary statistics were calculated based on R2 and RMSE from the validation sites associated with different spiking strategies and then compared with the model performance without spiking. The spiking test was carried out only on sites categorized under grassland according to the NLCD 2019 data layer, which accounts for about 25% of the full dataset (Table 3).

Estimate and evaluate field-scale soil moisture for Colorado sites

We used two sites in Colorado with available in-situ measurements as examples for generating daily, gap-free, high resolution, and multi-depth soil moisture layers. Although our codes were written in a way to allow user-defined area of interest (AOI) and spatial resolution, we set the AOI to our test sites and used a spatial resolution of 30 m to allow case study-based examination of modeling results and computational resources.

The first case study is the Cedaredge ranch, which has an area of 13.1 km2 and is located in Western Colorado (Fig. 3). The central and the southwest portion of the ranch property are dominated by irrigated pasture, where there is a mixture of cool-season perennial grasses (tall fescue, smooth brome, and orchard grass), white and red clovers, and alfalfa. The northeast portion of the property is predominantly open canopy forest while the rest of the ranch consists of a mixture of grass, legumes (alfalfa, clover), and shrubs. For this case study, we simulated soil moisture at a median depth of 7.5 cm to represent the 0–15 cm depth layer, and at a median depth of 80 cm to represent the 60–100 cm depth layer. Soil moisture was simulated for the year 2021 in order to compare against measurements from field campaigns carried out during the growing season of the year which aimed to capture field scale spatial variability. During the sampling campaigns carried out in May (5/24–5/26) and July (7/20–7/21) in 2021, soil moisture was determined for 41 sampling locations across the irrigated portion of the site for the 0–15 cm depth (Fig. S1) using the gravimetric method (Eq. (5)) based on soil cores (D = 5.25 cm) collected with field core samplers. Model estimates of 0–15 cm depth were also generated for the year 2019 due to the slight difference in model input source between 2019 and 2021 caused by data availability (Table 2).

Figure 3 Location of the study sites in Colorado and the associated high-resolution National Agriculture Imagery Program (NAIP) imagery of 2019.

The irrigated boundary is shown for the Cedaredge ranch and soil sensor locations are shown in the Central Plains Experimental Range (CPER) site.

(5) Gravimetricsoilmoisture(%)=Weightofwetsoil(g)−Weightofdrysoil(g)Weightofdrysoil(g)×100%

We used the Central Plains Experimental Range (CPER), which has an area of 92.8 km2 and is located in north-central Colorado, as the second case study (Fig. 3). The CPER site is dominated by rangeland plant communities including blue grama, plains prickly pear, sand dropseed, red three-awn, western wheatgrass, and four-wing saltbush (Steinert & Morgan, 2016). Since the site is included in the National Ecological Observatory Network (NEON) network, sensor-based soil moisture and temperature measures are available for public use (Metzger et al., 2019). We downloaded data from the Ameriflux data portal (https://ameriflux.lbl.gov/sites/siteinfo/US-xCP) before processing them to daily resolution and excluding frozen soils in R. Five sensors were installed approximately 30 m apart at the CPER site (Fig. 3) with measurements mostly available from 2019 and 2020 at multiple depths (Table S5). However, the data was not continuous in time due to sensor calibration requirements. In addition, the sensors covered different depth layers. Therefore, we retained observations from three depth layers (7, 46, and 66 cm) that had the most overlap among the five sensors for further analysis. Temporal trends of soil moisture were examined for 2019 with sensor averages during the time period when continuous measurements were available, mostly from the growing season to winter, for each depth layer.

For both case studies, covariates including soil properties, topography, and NLCD-based LULC classes that are used for model building were extracted from GEE with a gap-filling algorithm being carried out to replace missing data using the nearest neighbors (Table S2). In Google Colaboratory, we generated both the NLDAS data layer and the layer containing climate and biotic covariates at a daily time step. For biotic covariates, MODIS-based bitmasks were imposed to mask out poor-quality pixels, which were then replaced with temporal averages. In the case where data is missing or unavailable on a daily basis, a record from the closest date was used for temporal gap-filling. This simple interpolation method on time-series data ensures fast extraction of covariates. However, more sophisticated spatio-temperal gap-filling approach using information of neighboring pixels and moving windows coupled with geostatistical regression may be used in the future to further improve the accuracy of data from the missing dates (Weiss et al., 2014; Siabi, Sanaeinejad & Ghahrama, 2022). The MODIS-based tree cover% and RAP-based LULC class% were extracted for the year of interest with the closest year being used when data of the specific year is unavailable. The layers containing extracted covariates were then combined and deposited to the Google Cloud Storage Bucket (GCS) (Google Inc., Menlo Park, CA, USA) to allow data import to an R virtual machine implemented with the Google Cloud Computing Platform (GCP) (Google Inc., Menlo Park, CA, USA). The final step was to apply the QRF model derived from the full dataset to predict soil moisture based on the defined date range, depth layers, and spatial resolution for the AOI. The modeled 30 m resolution soil moisture estimates at the two sites were examined for their spatial and temporal patterns at the defined depth layers.

Results

Model performance influenced by calibration datasets and strategy

The model built upon the full dataset had a better fit with R2 = 0.53, RMSE = 0.078 m3/m3, RPD = 1.43) compared to the regionalized dataset that only contained Western U.S. records (R2 = 0.41, RMSE = 0.096 m3/m3, RPD = 1.30). The model performance was improved when only the top 0–5 cm layer soils were considered (full model: R2 = 0.58, RPD = 1.53; regionalized model: R2 = 0.53, RPD = 1.44) (Fig. 4). Model predicted soil moisture generally had narrower ranges than the observed values. Our model which encompassed a number of soil, climate, biotic, and topographic covariates as inputs outperformed those built only on NLDAS-derived soil moisture (NLDAS-SM), which showed model fits of R2 = 0.31 and R2 = 0.29, respectively, using the full and regionalized datasets for all the depths (Fig. S2). The NLDAS-only model also had lower predictive power than ours for the 0–5 cm depth (full model: R2 = 0.43; regionalized model: R2 = 0.42). The models built upon reduced covariate datasets had performance varying from very low (R2 = 0.33, RPD = 1.14) using only NLDAS-SM, sampling depth, LULC and remote sensing covariates, to close to the full model (R2 = 0.52, RPD = 1.43) when the top 15 most influential covariates are retained (Table S4). Our comparison illustrated that estimates of NLDAS-SM can be improved by adding modeling covariates but the covariates may be selected based on their variable importance to reduce computational costs.

Figure 4 Soil moisture model performance reported as error metrics according to the independent validation sites.

The model performance were shown for (A) all depths and (B) surface 5 cm depth samples of the full dataset and for (C) all depths and (D) surface 5 cm depth samples of the regionalized dataset. The full dataset contains observations from the Midwestern and Western U.S. states while the regionalized dataset only contains observations from the Western U.S. states due to the use of covariates from the rangeland analysis platform data layers.

The model performance decreased greatly with sampling depth (Fig. 5A and Fig. S3A) and was the greatest for forest and pasture (R2 > 0.5, RPD > 1.3), followed by grassland and cropland (R2 > 0.4, RPD > 1.25), and was the lowest for shrublands (R2 = 0.25, RPD = 1.2). Slight decrease in model performance (R2 between 0.4 and 0.45) was suggested for the winter months compared to the growing season (R2 > 0.5). This might be because there is generally a 60% decrease in data availability in the calibration dataset during winter than the rest of the year (Table 3). However, model performance evaluated for the winter months is associated with a larger variance than those reported for the growing season (Fig. 5C).

Figure 5 The model Coefficient of Determination (R2) and Root Mean Square Error (RMSE) derived from independent validation.

The soil moisture models built for different (A) sampling depths, (B) land cover types, and (C) sampling months. Model performance is shown for soil sampling depth up to 100 cm for (B) and (C). The full dataset containing observations from the Midwestern and Western U.S. states was used to build the calibration models. The model performance was presented as mean ± standard deviation based on five model runs.

The model performance represented by R2 had a range of 0.05–0.61 with a mean of R2 = 0.36 for grassland sites used for independent validation (Fig. 6A). Using data measured from 10% of the years to spike the national model and the remaining 90% of the years for validation, model performance was improved by 37% on average. Using 30% of the years for spiking and 70% of the years for validation further improved model predictive power where an averaged performance was calculated to be R2 = 0.53 with a range of 0.24 to 0.78. In addition to a more than 50% increase in model R2, this spiking strategy decreased model RMSE to 0.053 m3/m3, which is approximately 40% lower compared to not using any spiking strategies (0.092 m3/m3; Fig. 6B). Model performance (averaged R2 = 0.56 and RMSE = 0.051 m3/m3) was slightly improved by dividing the dataset evenly into spiking and validation sets for the study years but did not improve further when more than 50% of the samples were used for spiking (Fig. 6).

Figure 6 Distribution of site-based (A) model Coefficient of Determination (R2) and (B) Root Mean Square Error (RMSE) derived from soil moisture models built under different spiking strategies.

The spiking models were built on combined national and local datasets. The grassland sites from the full dataset which contains soil moisture observations from the Western and Midwestern U.S. states were used as the national dataset, which was used to build the model without local spiking. The local dataset was then randomly selected from 10%, 30%, 50%, 70%, and 90% of the years with measurements to spike the national model. Validation results were derived from comparing observed and modeled soil moisture for the rest of the years which were not used for model building at the site. The model performance was presented as mean ± standard deviation based on error metrics calculated for individual sites.

Variable importance of calibration models

Both the full and the regional models ranked soil texture (sand and clay contents), BD, SOC, NLDAS-SM, sampling depth, and elevation as the most influential covariates (Figs. 7A and 7B). This is consistent with the correlations computed between observed soil moisture and modeling covariates showing the strongest predictors as NLDAS-SM, soil properties, and elevation (Table S3). LULC-based covariates were in general less influential than soil covariates but more influential than RS-derived indices. Compared to the model built on the full dataset, the model built on the regionalized dataset ranked NLCD-derived LULC with slightly lower importance (Fig. 7B) probably because that information overlapped with the RAP-derived LULC class%. However, ranking between NLCD- and RAP-derived LULC vary among class-based models shown in Fig. S4 and this might be associated with certain LULC classes being more explanatory of the moisture variance within the category. The climate covariates generally ranked in the last one thirds in terms of their importance in building the QRF models regardless of their high temporal variability.

Figure 7 Variable importance for (A) the full dataset and (B) the regionalized dataset ranked according to the increase in node purity of the Quantile Random Forest model.

The full dataset contains observations from the Midwestern and Western U.S. states while the regionalized dataset only contains observations from the Western U.S. states due to the use of covariates from the rangeland analysis platform (RAP) data layers. The covariates include soil sand (Sand) and clay (Clay) contents, soil bulk density (BD), soil organic carbon (SOC), sampling depth (depth), NLDAS-derived soil moisture (SM), precipitation (ppt), temperature (T), vapor pressure deficit (VPD), Gross Primary Productivity (GPP), Enhanced Vegetation Index (EVI), Modis-based tree cover% (Tree%), land use land cover (LULC), Land Surface Temperature (LST), Normalized Difference wetness Index (NDWI), elevation (EL), slope (SL), aspect (AS), mean (mcurv), vertical (vcurv), and horizontal (hcurv) curvatures, Topographic wetness Index (TWI), surface roughness (SRG), and RAP-based estimates of annual herbs% (AFGC%), perennial herbs (PFGC%), bare ground (BG%), tree (TREE%), litter (LTR%), and shrub (SHB%) covers. Model performance was presented in the panels for comparison.

Elevation and NLDAS-SM were determined to be most influential for moisture model built on soils from the surface sampling depth (5 cm) (Fig. 8A). Besides NLDAS-SM and elevation, soil texture was also determined to have a predominant influence on soil moisture estimated from the deeper depth (100 cm) (Fig. 8B). Interestingly, sand% was ranked with higher importance than clay% in the 5 cm model while clay% was ranked as the most important covariate for the 100 cm model. This might illustrate the difference in water drainage and storage as controlling factors at different soil depths. The RS-derived variables such as LST and GPP, together with RAP-based estimates of LULC class percentages, were ranked with higher importance for surface compared to the deeper depth soil moisture models. When restricting the training data to the regionalized dataset, NLCD-derived general LULC classes were more influential than RAP classes for the deeper depth (Fig. S4B). Topographical covariates which reflect spatial rather than temporal differences were generally estimated to have higher importance in the deeper depth soil moisture modeling.

Figure 8 Variable importance ranked for soil moisture model built on soils from (A) 5 cm sampling depth, (B) 100 cm sampling depth, (C) grassland, (D) cropland, (E) January, and (F) July.

The full calibration dataset containing soil moisture observations from the Western and Midwestern U.S. states was used to build the models and the variable importance was reported based on the increase in node purity of the Quantile Random Forest model of the calibration dataset. The covariates include soil sand (Sand) and clay (Clay) contents, soil bulk density (BD), soil organic carbon (SOC), sampling depth (depth), NLDAS-derived soil moisture (SM), precipitation (ppt), temperature (T), vapor pressure deficit (VPD), Gross Primary Productivity (GPP), Enhanced Vegetation Index (EVI), Modis-based tree cover% (Tree%), land use land cover (LULC), Land Surface Temperature (LST), Normalized Difference wetness Index (NDWI), elevation (EL), slope (SL), aspect (AS), mean (mcurv), vertical (vcurv), and horizontal (hcurv) curvatures, Topographic wetness Index (TWI), and surface roughness (SRG). The model performance labeled in the panels were calculated as averages based on five independent model runs.

Variable importance was comparable for models built on soils from grasslands (Fig. 8C) and croplands (Fig. 8D) with soil properties, NLDAS-SM, and elevation ranked the highest, while climate variables were considered to be less influential. When the model is restricted to the grassland category which is most relevant to ranch managers, the most influential RAP estimates were perennial herbaceous cover (Fig. S4C). Both models built for January (winter) and July (summer) identified soil properties, NLDAS-SM, elevation, and tree cover to be the most influential covariates while precipitation was ranked with low importance (Figs. 8E and 8F). Sampling depth was considered to have higher importance in summer than winter which is associated with the seasonal variation of soil moisture of different depth layers.

Model application for case studies: time and resources needed for implementation

Model predictions for daily soil moisture were made at a spatial resolution of 30 m for the Cedaredge ranch and the CPER site and can be flexibly adapted by future users for their AOI and defined time frame using the following steps: (1) extraction of temporally constant soil and topographic covariates, (2) extraction of temporally dynamic climate and biotic covariates, (3) extraction of NLDAS soil moisture layers, (4) generate estimates, and (5) quality checking. The total time required for carrying out all steps for the Cedaredge ranch was about 5 h (Table S6) using a standard laptop computing machine with eight cores and 32 GB RAM. The computation time can be shortened if multiple steps, such as (2) and (3) were carried out simultaneously or if a virtual machine with a higher number of cores was set up to speed up the most time-consuming step (4). The total storage required for completing the task was about 150 MB. For the CPER site which is about 8 times larger than the Cedaredge site, the computation time was about twice (9.5 h) and the storage requirement tripled.

Evaluation of model estimates for case studies

The model estimated soil moisture, when averaged at the site level, was much more temporally dynamic for the top 0–15 cm compared to the deeper layer (60–100 cm), especially during the growing season for the Cedaredge ranch (Fig. 9A). The temporal trends were modeled to be more similar between the top and deeper soil depth layers for CPER (Fig. 9B) that is likely associated with differences in soil properties at these two sites. The spikes of top layer soil moisture typically matched with site-level precipitation patterns and when irrigation occurs during the growing season. Winter soil moisture was less dynamic and more representative of saturated levels. The averaged soil moisture level was modeled to be higher during the growing season in the deeper depth than the surface depth for both sites. Temporal dynamics of soil moisture were comparable at the site level regardless of the calibration datasets being used (Fig. S5). Spatially, the correlation between model estimated top layer (0–15 cm) soil moisture and that measured through the gravimetric method at the Cedaredge ranch during the growing season was significant but only moderate (R = 0.4, P < 0.01), with the measured soil moisture at the higher end (>20 m3/m3) being underestimated (Fig. S1). This might be due to the fact that soil moisture samples were taken across the irrigated portion of the ranch, but moisture level influenced by irrigation and grazing activities are mostly reflective from the RS-based inputs in the model that are only ranked with medium importance in the model (Fig. 7A). Moreover, the model estimates generated at the 30 m resolution would have a scale mismatch with the field samples taken at the point scale.

Figure 9 Soil moisture modeled for the year 2021 at the 0–15 cm and 60–100 cm depth layers in relation to precipitation.

The modeling results are shown at the Colorado (A) Cedaredge ranch and (B) the Central Plains Experimental Range (CPER) site. Precipitation data is shown in black boxes while the colored areas represent site-level standard deviation derived from soil moisture model predictions.

A wide range of correlations between modeled and measured temporal soil moisture (R between 0.28 and 0.57) with a mean of R = 0.42 (Table S5) was observed from the five sensors installed in CPER, which is independent of the broad-scale calibration dataset. Temporal validation at the site showed R = 0.64 for the top depth (7 cm) when sensor measurements were averaged (Fig. 10A). Noticeable mismatches in magnitude were observed in early July and mid-September where model estimates were lower than measured. Since the sudden spikes in measured soil moisture did not seem to be solely associated with precipitation events, they might be caused by management factors such as irrigation that are difficult to capture with the current model input. The correlation was observed to be moderate for the medium depth (46 cm: R = 0.87) and lowest for the deep depth (66 cm: R = 0.37). However, it should be mentioned that the magnitude of soil moisture was estimated to be consistently lower with model prediction compared to field measures. Moreover, sensor-based variation was observed to be much wider than modeled at deeper soil depths (Figs. 10B and 10C) which reflects the difference in footprint size for sensor measurements and model estimates.

Figure 10 Comparison of modeled and sensor measured soil moisture at three different depths including (A) top (7 cm), (B) medium (46 cm), and (C) bottom (66 cm) for the Central Plains Experimental Range (CPER) site.

The simulation was carried out in 2019 in order to compare to available sensor-based measurements. The area colored in grey represents sensor-based standard deviation when data is available. Standard deviation information is lacking for the 46 cm depth before September due to a lack of moisture records from multiple sensors. The area colored in red represents the model estimated standard deviation of 30 m buffered zones associated with the sensor locations. Pearson correlation for the temporal trends between modeled and sensor measured soil moisture was calculated and presented in the label. Precipitation f/or the investigated period was presented as black boxes.

Even though NLDAS-SM, which is used as the underlying dataset for downscaling the results, only consisted of two pixels for the extent of both of the case studies and therefore could not mirror spatially-explicit moisture dynamics (Figs. S6 and S7), the spatial pattern for model estimated soil moisture was able to illustrate much more spatially-refined patterns including the summer drying period for the irrigated pasture area of the Cedaredge ranch (Fig. S6). For the CPER site, the underlying NLDAS-SM caused a sudden change in moisture estimates within neighboring pixels (Fig. S7).

Discussion

Depth-based model performance compared with previous work

Previous work found RS-based soil moisture estimates to be moderately correlated with field-measured soil moisture for the surface but not deeper depth layers (Velpuri, Senay & Morisette, 2016). This aligned with validation results by depth shown in Fig. 5A. Our model performed similarly or slightly better (R2 between 0.5 and 0.6, RMSE between 0.07 and 0.075 m3/m3) than previous data fusion-based 0–5 cm soil moisture estimates that used independent sites as the validation set. The 12 studies that adopted independent validation scheme summarized in Table S1 reported a model fit (R2) between 0.22 and 0.66, with 58% of the studies reporting R2 ≤ 0.5 for the best-fitted model and 33% of the studies reporting an R2 between 0.5 and 0.6 like ours. The RMSE reported from these studies generally fell within the range of 6–8% which is comparable to our study. Jing, Zhang & Zhao (2018) achieved better performance (R2 = 0.66) but predicted soil moisture at coarser spatial (0.05°) and temporal (monthly) resolutions. The other studies which outperformed ours used alternative model calibration and validation strategies (Table S1). For example, random division of calibration and validation datasets was adopted by studies that reported more promising model fits (R2 > 0.8) (Liu et al., 2020; Greifeneder, Notarnicola & Wagner, 2021; Zhang et al., 2021). It should be noted that validation against randomly selected data points, even if stratified by study years, can generate much better model fits compared to independent sites because of the use of autocorrelated time-series data for both model training and validation (Meyer et al., 2018; Ploton et al., 2020). Our model fit for the validation dataset was calculated to be R2 > 0.95 (data not shown) if the random division method were to be adopted. However, such model validation schemes cannot guarantee model performance against independent sites which may differ significantly from the calibration dataset.

Karthikeyan & Mishra (2021) reported model performance for soil moisture estimated up to 100 cm and their conclusions are akin to ours showing a drastic decrease in model performance for the 50 and 100 cm depths (R2 < 0.3) compared to surface layers (R2 between 0.4 and 0.5). Our model outperformed theirs slightly but still had relatively poor fits for deeper depth layers (R2 = 0.29 for 100 cm and R2 = 0.40 for 50 cm, Fig. 5A) which cautions the use of soil moisture estimates at the root zone relying solely on the ML-based method. Deeper depth soil moisture tends to vary more by soil hydraulic properties, soil texture, and LULC compared to surface layers (Xiao et al., 2014; Karandish & Šimůnek, 2016; Das & Mohanty, 2006; Li et al., 2019). Even though our model did include LULC, LSM-derived estimates, and soil properties as covariates which were ranked with moderate to high importance for deeper soil layers (Fig. 7B), soil hydraulic properties were not directly included except that they could be inferred from soil texture, BD, and SOC (Saxton & Rawls, 2006). In the future, soil hydraulic maps (Hengl & Gupta, 2019) may be used to improve the modeling of soil moisture at deeper depths. Additionally, more adequate delineation of soil moisture variability of deeper depths may rely on techniques such as vadose zone and root growth modeling (Das & Mohanty, 2006; Yadav, Mathur & Siebel, 2009) that can better account for mechanisms associated with soil water flow and plant water uptake.

Model performance influenced by land cover and season

The model predictive powers for soil moisture were moderate for grassland, cropland, forest, and pasture (R2 > 0.4, Fig. 5B). By considering all depth layers, forest had the best prediction (R2 = 0.56, RMSE = 0.068 m3/m3) among the LULC types in our work, which is at odds with the study of Tavakol et al. (2019) that reported the highest error in estimating forest soil moisture using RS-based method because of the difficulty posed by a higher intensity of vegetation. The reason for this might be that NLDAS-SM correlated moderately (R = 0.65, Table S3) with observed soil moisture for forest and that our model incorporated soil and climate information in addition to the RS-based indices that are more sensitive to canopy covers. Moreover, the RMSE showed a very high variation among model runs using different sets of independent validation sites, meaning that model accuracy can vary greatly with site factors for forest prediction (Fig. 5B). The model fit was better for grassland (R2 = 0.45, RMSE = 0.088 m3/m3) than cropland (R2 = 0.4, RMSE = 0.103 m3/m3) in our study, which is in line with the finding that SMAP-downscaled products generated lower RMSE for grassland due to its homogeneity compared to cropland (Wu et al., 2015). Unfortunately, model performance was relatively poor for shrublands and this could explain that the regionalized model using RAP covariates did not outperform the full model probably because of the higher proportion of sites being categorized under shrubs (36%) for the Western U.S. (Table 3). It could be that the average soil moisture contents are much lower in shrublands than in others and so the variation within a smaller data range was harder to capture. However, mixed results were found in other studies comparing model performance of shrublands with other LULC types (Liu et al., 2020; Senyurek et al., 2020; Fuentes, Padarian & Vervoort, 2022). The difference in soil moisture dynamics tied to LULC types could be explained by vegetation-associated evaporation and precipitation infiltration, soil BD and rooting depth, and initial soil moisture contents (Niu, Musa & Liu, 2015). Building models to better refine soil moisture estimates for diversified systems under various LULC types may therefore consider incorporating additional soil and biotic factors in the future.

Our validation showed slightly better model performance during the growing season than winter months (Fig. 5C). The greatest proportion of training data was taken from June to September (Table 3), which coincided with the slightly better agreement between the observed and modeled soil moisture. The model performance was lower (R2 ≤ 0.4) with high variance between model runs in January and February, but RMSE was also low (<0.045 m3/m3) because of the lower temporal variation in soil moisture during this time period. Even though RS-based indices were found to have good correlations with soil moisture (R ranged between 0.5 and 0.7) for a winter wheat field (Ren et al., 2022), our model did not emphasize spectral indices as influential covariates (Fig. 8E) probably due to the complexity of LULC and depth classes involved in the training dataset. Winter soil moisture may be better simulated with process-based models that can account for effective pore spaces, heat exchange, and evaporation during the freeze and thaw processes (Kahimba, Ranjan & Mann, 2009; Banimahd & Zand-Parsa, 2013).

Covariate importance for empirical estimates of soil moisture

Soil properties including soil texture, BD, and SOC were consistently identified to be among the most influential covariates for building the ML-based soil moisture models (Figs. 7 and 8). This is not surprising because soil properties, especially soil texture and pore size distribution, are highly influential on soil hydraulic properties (Saxton & Rawls, 2006; Strudley, Green & Ascough, 2008; Cichota et al., 2013; Taghizadehghasab, Safadoust & Mosaddeghi, 2021), which in turn determine lateral and horizontal water movements. On one hand, process-based models such as Century (Parton & Rasmussen, 1994) and EPIC (Izaurralde et al., 2006) have commonly employed pedo-transfer functions to estimate water holding capacity which is then used to estimate soil moisture dynamics through the simple bucket model. On the other hand, the empirical method which is being increasingly used to model or downscale soil moisture has largely relied on DEMs and RS products (Table S1). Recent advances in DSMs and global soil monitoring networks (e.g., GlobalSoilMap) (Hartemink et al., 2010; Chen et al., 2022) should enable the use of gridded, quality-controlled products of soil properties as inputs for both process-based and empirical models.

The NLDAS-derived estimates of soil moisture and soil sampling depth were also ranked with very high importance for all the models (Figs. 7 and 8). Based on variable importance, it might be suggested that the NLDAS data product would be suited for studies carried out at the broad scale where fine-scale soil moisture pattern is not required to drive the modeling of spatial dynamics. This notion is supported by the fact that the direct correlation between NLDAS and measured soil moisture is moderate (R = 0.5) for all layers and good (R = 0.7) for the top 0–5 cm layer (Table S3). Soil sampling depth could be used either as a covariate like in this study or as a class factor to develop separate models. Considering that models built for surface and deeper depth layer soil moisture estimates had relatively different rankings in covariate importance (Figs. 8A and 8B), especially with regards to RS-derived covariates, it may be advisable to build separate models for different depths layers by selecting customized sets of covariates.

Elevation was ranked with high importance for all models (Figs. 7 and 8), which mainly reflects site-based differences. In contrast, other DEM-derived covariates such as aspect and curvatures had lower rankings in many instances. When extracting soil moisture data points from national databases, only one coordinate is provided for each site, meaning that field-level spatial variation could not be accounted for. This could explain the discrepancy that topographic covariates are often identified with high importance for field-scale soil moisture estimates (Zhu & Lin, 2011; Lee & Kim, 2019) but not in this study. One way to compensate for the limitation of applying a broad-scale model to the fine spatial scale where there is a lack of topographic explanation on soil moisture is to spike the broad scale model with higher resolution soil moisture maps using topographic covariates. These maps can be derived from field sampling campaigns or well-validated, RS-based data fusion products that focus on delineating spatial rather than temporal soil moisture dynamics. However, it remains to be tested whether the high-resolution maps need to be highly representative of the condition that is similar to the target site in order to make the model prediction successful.

The RS-derived indices exhibit both spatial and temporal variations but were ranked only with medium importance when all records are considered (Fig. 7B). Furthermore, RS indices had relatively weak correlation with observed soil moisture (Table S3), which, combined with our finding that model excluding RS covariates performed similarly to the full model (Table S4), suggests that future modeling work might retain only the most influential RS covariates to reduce computation time and risks of model fitting. It could be that some of the variance provided by RS indices overlapped with those from NLDAS-SM and site-based soil properties. Nevertheless, modeling of surface soil moisture relied more on GPP and LST (Fig. 8A), which makes sense as GPP is closely tied to plant productivity that is known to be influenced by soil moisture conditions (He et al., 2016; Fu et al., 2020) and LST derived from the thermal bands informs soil moisture status through its association with energy balance and hydrology (Sandholt, Rasmussen & Andersen, 2002; Cammalleri & Vogt, 2015). For deeper depth, soil moisture appeared to be best correlated with GPP and tree cover among the RS indices (Table S4), which demonstrated the association between soil moisture and belowground biomass that can be inferred from surface indices. The RAP and NLCD-based LULC covariates were also frequently ranked with medium importance (Fig. 8 and Fig. S4). RAP provided more refined temporal and class percentage information compared to NLCD and so was ranked slightly higher than NLCD in many cases. However, the RAP product is especially oriented toward the rangeland ecosystem and does not include detailed information or threshold values for croplands or different types of forests. It may therefore be advisable to build soil moisture models with combined use of NLCD and RAP for areas with more mixed LULC types.

Climate covariates that reflect both site-based and temporal differences were generally ranked with low importance (Figs. 7 and 8). In view of the feedback between precipitation and soil moisture, and the fact that the former is a main driver of the latter (Salvucci, Saleem & Kaufmann, 2002; Sehler et al., 2019), it was a bit surprising that precipitation was consistently ranked with lower importance and had poor correlations with observed soil moisture regardless of LULC and time of the year (Table S4). The cause for this might be that there is usually a time lag between rainfall events and changes in soil moisture, in which case the anomalies of soil moisture and precipitation might be better correlated, and an antecedent precipitation index may be used instead of daily precipitation (Wagner, 2003; Koster et al., 2004; Schoener & Stone, 2020). Another explanation is that the climate data have relatively coarse spatial resolution compared to other covariate datasets (Table 2), which would more likely cause a spatial mismatch between the measured data point and the value extracted from the coarse-resolution pixel.

Model spiking and evaluation of soil moisture estimates for grassland sites

As illustrated previously, the drastic difference between model performance caused by the validation scheme (random sample selection vs independent sites) demonstrated relatively poor transferability of the soil moisture model from the broad to the fine spatial scale. The spiking test which was meant to alleviate the scale mismatch for model training showed the need to incorporate at least 30% of the local measurements into the national model in order to reduce RMSE to be lower than 0.05 m3/m3 (Fig. 6B), which means that site-based soil moisture dynamics might need to be monitored for three years within a ten-year time window for the grassland sites. That being said, increasing spiking samples to more than half of the study years did not further improve model performance (Figs. 6A and 6B), indicating potential overfitting of temporal soil moisture dynamics that can vary year by year.

The soil moisture estimated for the Cedaredge ranch showed reasonable temporal dynamics (Fig. 9) with winter moisture levels estimated to be close to field capacity due to snowmelt and reduced evapotranspiration, which is followed by a drying period during the growing season especially at the surface depth (Moran et al., 2000; James et al., 2003). The soil moisture estimated for the CPER site demonstrated reasonable temporal matches with sensor measurements for the top 0–15 cm depth in terms of magnitude and direction of change (R = 0.64, Fig. 10). However, some of the spikes observed by the sensors, especially those with >20% change within a week, seem to have been missed or reduced in model simulations (Fig. 10A). The spikes associated with precipitation were better captured which makes sense as climate data was incorporated into the model. Other spikes caused by management practices such as irrigation and grazing are likely harder to model and are mostly reflected by RS-derived indices which are sometimes subject to temporal mismatches and data quality issues. One way to improve the modeling of soil moisture responses to management would be to set up controlled experiments with contrasting management practices in order to explore the explanatory covariates for soil moisture dynamics within a short period of time. The sensor readings also indicated higher temporal and spatial variations for deeper depth layers which were not mirrored by the modeling results (Figs. 10B and 10C). The modeled temporal changes in deeper depth soil moisture may be enhanced by adding covariates reflecting soil hydraulic properties and management. Since the sensors were relatively close to each other in the CPER site (Fig. 3B), a lack of difference among sensors from the model simulation is likely related to the input data not being able to represent refined spatial patterns.

The correlation between modeled and measured soil moisture at 0–15 cm depth during the growing season was found to be only moderate (R = 0.4, RMSE = 0.036 m3/m3) when data from one sampling campaign focusing on spatial variability was used (Fig. S1). The discrepancy is associated with a narrower range of soil moisture predicted compared to captured, which likely reflects the difference in modeled resolution (30 m) and point-scale measurements together with the difficulty to track irrigation-associated change in soil moisture. Additionally, some model inputs such as soil and climate data as well as RS-indices have coarser resolution ( ≥100 m) which can lead to a reduced modeling power at fine resolution prediction. Although modeled soil moisture was able to delineate summer drying clustered within the irrigated area in the Cedaredge ranch compared to the northeastern open canopy area which had smaller moisture change from winter to summer, the contrast between LULC was likely exaggerated in winter (Fig. S6). One explanation might be that the winter calibration data was relatively small (Table 3) which made it harder to build more robust and transferable predictions. Another explanation might be associated with the fact that the soil moisture estimates were strongly influenced by the underlying soil maps as evidenced by covariate ranking. Spatial pattern simulated at the CPER site reflected similar patches of high and low soil moisture estimates for summer and winter but was much more restricted to the tile difference of NLDAS-SM (Fig. S7). In this case, interpretation based on field averages or relative differences among pixels within the same NLDAS tile might be more reliable than the interpretation of soil moisture magnitude at the whole site based on modeled spatial patterns. The relatively consistent magnitude of standard deviation in soil moisture estimated from the whole ranch (Fig. 9) reflected the notion of temporal persistence of soil moisture changes in space (Zucco et al., 2014).

Implications, limitations, and next steps

Our work provided a protocol for building ML models at the broad scale before applying them to the AOI with user-defined resolutions (Fig. 2). Even though building a national model using long-term soil moisture records can be time-consuming given the size of the calibration dataset, the actual computation time needed to generate site-based, multi-depth, 30 m, daily soil moisture layers for our case study took less than 10 h for a ~100 km2 ranch by utilizing HPC (Table S6), which makes it realistic for users with basic model training to obtain field-scale moisture estimates needed for monitoring purposes or as model inputs. We made our model and codes open source (Table S2) so that users may choose to apply or adapt our model to make predictions for their AOI at the defined time and depths. Additional covariates or those from alternative data sources can be used to rebuild models like ours.

Several alternative soil moisture products derived from LSM and/or RS data, which were used by previous studies (Table S1), such as the Soil Moisture Active Passive (SMAP) (Entekhabi et al., 2010), Global Land Data Assimilation System (GLDAS) (Rodell et al., 2004) and Climate Change Initiative (CCI) (Dorigo et al., 2017), may be used as an alternative to NLDAS for model training. In our study, we selected NLDAS as a baseline soil moisture product for downscaling (i.e., used as a covariate) because it has a much wider temporal coverage (since 1979) compared to SMAP (since 2015), and is generated at finer spatial resolution (0.125°) than GLDAS (1°) or CCI (0.25°). Furthermore, although GLDAS and USDA-enhanced SMAP are deposited in GEE and more readily available for HPC users, the NLDAS data layers can be downloaded and processed in HPC to allow both broad-scale and fine-scale data extraction (Table S2). The various soil moisture products have been thoroughly validated against in-situ measurements and compared against each other (Ford & Quiring, 2019; Deng et al., 2020; Jiang et al., 2020). However, future work is needed to generate long-term, fine spatial resolution soil moisture products that would be beneficial to users focusing on field-scale research or management. Currently, users would have to choose between using publicly available but coarser resolution soil moisture datasets or developing their own products which can be costly and time-consuming.

In addition to using alternative soil moisture products as baseline maps, covariates that were not included in this study should be examined in terms of their potential to improve model performance and transferability. For example, the shortwave infrared band (SWIR) which is available from most remote sensors is proposed to be useful for improving soil moisture estimates (Sadeghi et al., 2017). A number of RS-derived indices, including Leaf Area Index (LAI), temperature vegetation condition index (TVDI), and crop water stress index (CWSI) (Patel et al., 2009; Sánchez et al., 2012; Chen, Willgoose & Saco, 2015; Akuraju, Ryu & George, 2021), have also been reported to be suitable covariates for soil moisture modeling. Other commonly used covariates for soil moisture modeling include albedo, solar radiation, potential evapotranspiration, and soil hydraulic properties (Table S1). A model selection procedure may be necessary to ensure model efficiency if model training involves a large number of covariates that are correlated.

The next step is to validate our modeling results at additional sites from the U.S. Western states. The ML-based model can perform reasonably well and is specifically useful for data-poor areas where field-measured parameters such as soil hydraulic properties needed to run process-based models are missing (Carranza et al., 2021). Nonetheless, the ML-based method is limited in transferability. Specifically, the controlling factors of soil moisture at the broad scale (e.g., national level) usually differ from those at the fine-scale (e.g., site level). One way to reduce the potential model overfitting or mismatch at the field level using broad-scale data for training is to adopt a localized calibration method by selecting the most representative calibration samples at the pixel level; however, that procedure would be extremely time-consuming and computationally intensive. A less costly approach would be to select appropriate calibration samples for each site from the national library and spike those with local sensor data. The spatial variation of estimated soil moisture at the field level using the full or a selected national library data combined with different number of local sensor measurements should therefore be compared and verified against field observations. Ideally, samples can be collected from different areas of the field to validate spatial patterns. However, such high spatial resolution validation may only be practical for several sampling campaigns within a year unless a substantial sensor network is established at the site level.

Field-based calibration and validation would be necessary when field characteristics do not fall within the domain of the calibration datasets and therefore cannot be estimated accurately based on observations from the national databases alone. The spiking method incorporating site-based temporal data can improve model transferability but would require site-based measurements. Using sensors installed at a lower spatial resolution, field-scale model calibration and validation can be carried out temporally such that in-situ measurements of soil moisture from different time periods can be used for model spiking and validation. In this case, it would be critical to select the optimal spiking dataset (Nawar & Mouazen, 2017) needed to achieve a balance between model accuracy and measurement costs. We generated spiking samples based on sampling years as suggested by Abowarda et al. (2021), but future work should investigate the use of data at a coarser temporal resolution or only select those that are representative of the seasons of interest in order to reduce monitoring costs.

Conclusions

This work provided a transparent method with open-source codes to generate daily, high-resolution soil moisture estimates by building machine learning models based on national soil moisture databases (SCAN and USCRN), LSM-derived coarse resolution soil moisture estimates (NLDAS-SM), and environmental covariates including soil properties, climate variables, DEM and RS-derived indices, and LULC-based classes. Among the covariates investigated, NLDAS-SM, soil properties, and DEM-derived indices were identified to be most influential.

The model performed similarly or better to other studies carried out for the contiguous U.S. using independent sites for validation, and greatly outperformed those using NLDAS-SM alone. Our model performance (R2 ≥ 0.5, RMSE < 0.078 m3/m3, RPD > 1.4) at 5 and 10 cm illustrated that the data fusion-based model is effective at simulating surface soil moisture. Field-scale validation showed acceptable matches between observed and modeled soil moisture both temporally (R = 0.64) and spatially (R = 0.4) at the top 0–15 cm depth layer. It was confirmed that the machine learning-based empirical model is relatively weak at predicting soil moisture from the 50 and 100 cm sampling depths, which calls for the use of additional hydraulic covariates or coupling of mechanistic models for simulating root zone soil moisture. Our study obtained moderate agreement between observed and simulated soil moisture for grassland and cropland but was much weaker at predicting moisture from shrublands which likely contributes to the reduced model performance using a regionalized dataset from the Western U.S. states. Winter soil moisture was predicted with less accuracy caused by a lack of training data and the challenge to account for the freeze-thaw events.

The relatively poor model transferability evidenced by the difference in model performance between random and independent validation emphasized the need to use techniques to reduce mismatch associated with applying broad-scale calibration models to the fine-scale. Spiking the national model with local samples significantly improved the model’s ability to predict temporal soil moisture patterns (R2 > 0.6 and RMSE < 0.05 m3/m3) but model accuracy for predicting spatial soil moisture at the site level needs to be further validated and improved with optimized calibration datasets.

Supplemental Information

Supplemental Information 1 All codes for this work.

Click here for additional data file.

Supplemental Information 2 Supplementary Figures and Tables.

Click here for additional data file.

We would like to acknowledge Dr. John Kimball for providing insight and expertise that greatly assisted with the method and data development. Special thanks to Dr. Jingyi Huang for providing open-source codes and suggestions which helped us improve the methodology. We appreciate the help from Dr. Edward Ayres in answering questions regarding the CPER sensor data. Our gratitude also extends to Jim Howell and Brandon Dalton for maintaining the Colorado site and examining our site-level modeling results. The communication and data exchange with Dr. Jennie DeMarco has helped to improve the contents of the manuscript. Research Assistant Haydee Hernandez helped with communications and data acquisition in Colorado.

Additional Information and Declarations

Competing Interests

Author Contributions

Data Availability

The authors declare that they have no competing interests.

Yushu Xia conceived and designed the experiments, performed the experiments, analyzed the data, prepared figures and/or tables, authored or reviewed drafts of the article, and approved the final draft.

Jennifer D. Watts conceived and designed the experiments, prepared figures and/or tables, authored or reviewed drafts of the article, and approved the final draft.

Megan B. Machmuller conceived and designed the experiments, authored or reviewed drafts of the article, and approved the final draft.

Jonathan Sanderman conceived and designed the experiments, prepared figures and/or tables, authored or reviewed drafts of the article, and approved the final draft.

The following information was supplied regarding data availability:

The codes used to generate the models in this study are available in the Supplemental File and at GitHub (https://github.com/xiayushu/RCTM-soil-moisture).

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
