# Peer review of "Machine learning based estimation of field-scale daily, high resolution, multi-depth soil moisture for the Western and Midwestern United States"

_PeerJ, doi:10.7717/peerj.14275_

## Round 0.1 · original submission · Minor Revisions

The results of high-resolution machine learning-based estimation of daily soil moisture for the western and midwestern United States are eligible for PeerJ submission. The article is written in high-quality English and is well structured. A consistent and logical presentation of the material contributes to the reader's perception of information. Also, the article is well filled with illustrative material that enhances its informativeness. The tested research algorithm made it possible to qualitatively reveal the essence of the problem of soil moisture assessment using machine learning methods.

The application of digital soil mapping is aimed at improving soil moisture management, as illustrated by the example of the US West and Midwest. Also, the research method chosen by the authors of the article and its description reveal the possibilities of using machine learning for the purpose of evaluating the improvement of soil moisture models taking into account spatio-temporal variables. Confirmation of causal relationships is reflected in the large-scale QRF model, the best performance of which was observed for forests and grasslands.

Calibration of data and a qualitative combination of research methods led to obtaining reasonable conclusions about the accuracy of the model for the upper depth of the soil. They are important for the further study of soil moisture in the region and contribute to the possibility of reducing the cost of monitoring studies. Considering these facts, I appeal to the authors of the article to react and take into account, if possible, the comments (wishes) of the reviewers who volunteered to review the article in order to improve its quality.

Reviewer 1 ·

Basic reporting

No comments

Experimental design

No comments

Validity of the findings

No comments

Additional comments

The use of color cards improved the perception of the presented material

Reviewer 2 ·

Basic reporting

This study used QRF to estimate soil moisture. This study has a certain innovation, compared with previous studies, it has improved the prediction accuracy of soil moisture to a certain extent. This paper can be accepted after correcting major comments.

Experimental design

1.Line 206-214, the authors chose so many covariates, did they consider the problem of multicollinearity among covariates? Some parameters are easy to be redundant, which leads to the degradation of the accuracy of machine learning models.
2.Figure 7.8 shows the node purity of QRF, showing the contribution degree of each factor (refers to the synthesis of information gain generated by this feature in multiple branches, and the contribution rate of all factors is equal to 1) can intuitively better express the importance degree of each covariate. In addition, some statistical analyses of the association of these covariates with soil moisture at different land uses and at different depths seem to be missing. It may be better to select more effective parameters according to different methods (Pearson correlation, Spearman correlation, mutual information method) for different types of land use.
3.line 284-285 How the Quantile random forest works mathematically? Can authors provide the Quantile random forest tree with their model?
4.line294-295, The RPD measurement values can be used to evaluate the accuracy.
5.Line 327, on which day was the sample taken? What tools? The gravimetric method formula should be given.
6.Why did the authors use the QRF model, and is it the best model among all machine learning models of Table S1?
7.line 356-357, What is the basis for temporal gap-filling?
8.line 587-588, the most important factors affecting soil hydraulic characteristics are usually soil texture and particle size distribution (Strudley et al., 2008). The determined particle size distribution and BD are usually also determined, while the soil matrix potential is the potential energy caused by soil solid and capillary force, which is a change value. The larger the negative value, the drier the soil. 0 when the soil is saturated (Taghizadehghasab et al., 2021).
Strudley, M.W., Green, T.R., Ascough, J.C., 2008. Tillage effects on soil hydraulic properties in space and time: State of the science. Soil and Tillage Research 99, 4-48.
Taghizadehghasab, A., Safadoust, A., Mosaddeghi, M.R., 2021. Effects of salinity and sodicity of water on friability of two texturally-different soils at different matric potentials. Soil and Tillage Research 209, 104950.

Validity of the findings

no comment

Additional comments

no comment

Reviewer 3 ·

Basic reporting

Overall, the authors did a good job laying out the background of the research, the experiment design and data collection of the research, and the use case of the research result. There is no noticeable grammar mistakes. The paper reads smoothly.

The literature review is extensive and thorough. The argument is convincing that there needs to be a model that incorporates a few factors to predict the soil moisture, for example, seasonal effect, ranch management, depth of the soil layers, etc. The contribution to the accurate prediction for deeper layers is especially important.

Experimental design

There are plenty of details to the experiment design and the data collection in the paper. There are sufficient explanations on where the sites are, the source of the data, and timeframe of the data collection. It is great to see that the authors carefully control the experiment by dividing the modeling processed into regions, layers and sites, etc.

Validity of the findings

The paper can benefit greatly from adding a few more sentences to the statistical analysis. Throughout the paper, it’s obvious that the authors did quite a few modeling experiments with a couple of R packages. However, only the training/testing data is explicitly explained. The reason of using QRF model, its formulas and the parameters are all missing from the paper. Please use a few sentences to help readers understand better.

---

## Round 0.2 · accepted · Accept

A high-resolution machine learning-based soil moisture study for the western and midwestern United States is complete and ready for journal publication. The authors qualitatively and thoughtfully took into account the recommendations of the reviewers, which contributed to a noticeable increase in the quality of the article.